# Hyperbolic metamaterial empowered controllable photonic Weyl nodal line semimetals

Shengyu Hu [1], Zhiwei Guo [1] ✉, Wenwei Liu[2,3,4], Shuqi Chen[2,3,4,5] & Hong Chen [1]

Motivated by unique topological semimetals in condensed matter physics, we propose an effective Hamiltonian with four degrees of freedom to describe evolutions of photonic double Weyl nodal line semimetals in one-dimensional hyper-crystals, which supports the energy bands translating or rotating independently in the form of Weyl quasiparticles. Especially, owing to the unit cells without inversion symmetry, a pair of reflection-phase singularities carrying opposite topological charges emerge near each nodal line, and result in a unique bilateral drumhead surface state. After reducing radiation leakages and absorption losses, these two singularities gather together gradually, and form a quasi-bound state in the continuum (quasi-BIC) ring at the nodal line ultimately. Our work not only reports the first realization of controllable photonics Weyl nodal line semimetals, establishes a bridge between two independent topological concepts–BICs and Weyl semimetals, but also heralds new possibilities for unconventional device applications, such as dual-mode schemes for highly sensitive sensing and switching.

Beyond Landau's classic approach based on spontaneously broken symmetries, topological theory provides a new perspective on the classification paradigm, including gapped phases like topological insulators[1], and gapless phases like topological semimetals[2]. During the past decade, topological semimetals have been widely discussed for emulating relativistic quasiparticles[3], for example, fourfold Dirac fermions[4,5] and even twofold Weyl fermions[6,7], which possess linear crossing bands with unique transport properties like delocalization[8], Zitterbewegung[9,10], and Klein tunneling[11,12]. Through breaking inversion symmetry and/or time-reversal symmetry, one Dirac fermion can split into a pair of Weyl fermions carrying opposite two-dimensional (2D) topological charges, which are defined by the distinguished Chern number[13,14]. Such transition establishes a template for the double Weyl semimetals[15]. Moreover, these gapless phases can be classified into nodal points, nodal lines, and nodal surfaces according to the dimensions of band degeneracy areas. For example, Gapless phases can proceed to transit from a pair of Weyl nodal points to a single Weyl nodal line after eliminating spin-orbital coupling[16] or introducing the time-reversal symmetry[17], and Nielsen-Ninomiya no-go theorem determines that such nodal lines tend to be globally trivial[18,19]. Local properties of one-dimensional (1D) topological charges are widely discussed hence: on the intersecting surface, any loop interlocking with/without the nodal line has π/0 Berry phase (owing to the $\mathbb{Z}_2$ class), and nontrivial Berry phase protects the nodal line[20] and drumhead surface state (DSS)[21] effectively. For multi-band configurations of nodal lines, like nodal chains[22] and nodal links[23,24], additional quaternion charges are determined by the geometric relation between different intersecting surfaces, and exhibit non-Abelian characteristics.

[1]MOE Key Laboratory of Advanced Micro-Structured Materials, School of Physics Science and Engineering, Tongji University, 200092 Shanghai, China. [2]The Key Laboratory of Weak Light Nonlinear Photonics, Ministry of Education, School of Physics and TEDA Institute of Applied Physics, Nankai University, 300071 Tianjin, China. [3]Renewable Energy Conversion and Storage Center, Nankai University, 300071 Tianjin, China. [4]Smart Sensing Interdisciplinary Science Center, Nankai University, 300071 Tianjin, China. [5]The Collaborative Innovation Center of Extreme Optics, Shanxi University, 030006 Taiyuan, Shanxi, China. ✉e-mail: 2014guozhiwei@tongji.edu.cn

Otherwise, recent interests are inspired in dual nodal rings, which transit from different Weyl nodal points respectively. Such dual-band configurations recover the discussion of real Chern number, and induce higher-order topological states[25,26]. As the basic elements among these configurations, single rings[27] or straight lines[28] construct the enormous territory of nodal physics. It would be highly desirable to excavate their evolution dynamics and new topological features.

In this paper, we propose simple structures as 1D photonic crystals (PCs) have, and they can provide a versatile platform to exhibit flexible phase transitions of double Weyl nodal rings (WNRs), which are composed of two uncoupled WNRs with different polarizations. After considering hyperbolic metamaterials (HMMs) with conductive sheets into PCs, we introduce additional degrees of freedom (DOFs) of rotation and translation into photonic Weyl quasiparticles, which could hardly coexist and be modulated independently in previous systems. Based on such a platform, some unprecedented topological properties of WNRs, which tend to be seen as relatively trivial for lack of complex topological structures, are further elucidated. Generically, the above-mentioned DSS resides unilaterally in the electronic[21,26] or photonic systems[5,27], corresponding to the region with nonzero Berry phase inside or outside the nodal ring. However, by breaking the inversion symmetry of unit cells here, a pair of reflection-phase singularities (related to the exceptional points, EPs) carrying opposite topological charges emerge near each WNR, and pin a unique bilateral DSS, which spans both the inner and outer regions of the WNR on the projected surface Brillouin zone. After reducing radiation leakages and absorption losses, the two singularities of bilateral DSS can gather together and form a scattering-matrix singularity (corresponding to a bound state in the continuum, BIC) at the WNR ultimately. Given that this result is widely available for the above semimetal phases of rotation or translation, our results first unveil DSSs become an intriguing bridge between two subfields of topological physics, Weyl semimetals and singularities (EPs and BICs), which have been rarely discussed before. Wherein ill-defined singularities include EPs and BICs under the framework of scattering theory. The EP denotes the coalescences of eigenstates, and the BIC is a unique mechanism with zero leakage and zero linewidth[29,30], and widely discussed with the structure containing in-plane anisotropic materials[31,32], or epsilon-near-zero materials[33,34] in the 1D layered system before. On the other hand, such Weyl quasiparticles possessing the properties of quasi-BICs can be manipulated through DOFs of rotation or translation, generating degenerate quasi-BICs in the phases of Dirac or quasi-Dirac nodal rings. At the same time, phase singularities possess analogous properties, and we propose a sensing strategy of dual modes based on Heaviside-like phase jumps near the singularities with two polarizations.

## Results
### Phase transitions of rotation and translation
Before delving into the details of the 1D PC, we focus on an intuitive theoretical model that reveals the intricate classifications between various Weyl nodal line semimetals (WNLSs). In the compound space composed of energy $E$ and momentum $q$, a nodal line can be seen as a 2D manifold containing infinity nodal points. Such properties exist naturally in complicated systems, including famous WNLSs and Dirac nodal line semimetals (DNLSs). Typically, the WNLSs can be described by a two-fold-degenerate Hamiltonian. With co-dimension 2 of nodal lines, it is necessary introducing two variables $q_a$ and $q_b$ to construct a subspace[35]. The effective Hamiltonian can take the form of $H_I = H(\Delta, \phi, q_1, q_2)$, where constants $\Delta$, $\phi$, $q_1$, and $q_2$ provide four modulation DOFs of the Weyl quasiparticle. Especially, the detune $\Delta$ denotes the initial energy of the Fermi surface (FS), which determines the position of the nodal line. The tilt angle $\phi$ controls the rotation of the cone-like band around the axis of $q_a = q_1$, which lies on the FS. Three regions of $\phi$, $(n\pi - \pi/4, n\pi + \pi/4)$, $(n\pi - \pi/4, n\pi + 3\pi/4)$, and

critical $n\pi \pm \pi/4 (n \in \mathbb{Z})$ differentiate the type-I, type-II and type-III WNLSs, as shown in Fig. 1a–c, respectively. Observing the band projection on the plane $E = \Delta$, type-I/II/III WNLSs have point-like/cross-like/line-like FSs, colored by red. It is notable that type-III WNLSs have a unique band edge mode, which supports zero group velocity. In the extended space $E - q$, the constants $q_1$ and $q_2$ are related to the variables $q_a$ and $q_b$, which have a decisive influence on the 2D route of nodal lines. In 1D photonic systems, a conversant configuration is the WNR, which $q_a$ and $q_b$ are chosen as $q_a = k_\rho = \sqrt{k_x^2 + k_y^2}$ and $q_b = k_z$, respectively. Here, $k_\rho$, $k_x$, $k_y$, and $k_z$ represent the wave vectors along the radial $x$, $y$, and $z$ directions[5]. As is shown in Fig. 1d–f, they can be deemed as the cone-like bands moving along the routes marked by white lines. And $q_1$ corresponds to the radius of the nodal ring. Typically, $q_1$ ($q_2$) describes the sizes of closed manifolds or locations of open manifolds for projections along $q_a$ ($q_b$). Subsequently, we focus on the situation of decoupled multi-quasiparticles, where double WNRs for different pseudospins $\alpha$ and $\beta$ coexist in the $E - q$ space. The effective Hamiltonian turns into: $H_{II} = H_I^\alpha (\Delta^\alpha, \phi^\alpha, q_1^\alpha, q_2^\alpha) \oplus H_I^\beta (\Delta^\beta, \phi^\beta, q_1^\beta, q_2^\beta)$. Focusing on the DOFs of location $(\Delta, q_1, q_2)$, double WNLSs can be classified into three steps: DNLSs ($\{\Delta^\alpha, q_1^\alpha, q_2^\alpha\} = \{\Delta^\beta, q_1^\beta, q_2^\beta\}$), quasi-DNLSs ($\Delta^\alpha = \Delta^\beta$, $\{q_1^\alpha, q_2^\alpha\} \neq \{q_1^\beta, q_2^\beta\}$) and other isolated WNLSs ($\{\Delta^\alpha, q_1^\alpha, q_2^\alpha\} \neq \{\Delta^\beta, q_1^\beta, q_2^\beta\}$) as shown in Fig. 1g–i respectively, which correspond to the evolutionary process of translation with the gradual separation in the dimension of momentum and energy. This effective Hamiltonian might be realized in the systems supporting two orthogonal modes, like polarizations in electromagnetic or elastic waves[36], or two different mechanisms, like electromagnetic and elastic waves[37,38].

### Photonic Weyl nodal line platform
Then we study the above-mentioned double WNLSs in the photonic system, where $q_a$ corresponds to the radial wave vector $k_\rho = k_0 \sin\theta$, while $q_b$ corresponds to the Bloch wave vector $k_z$. $k_0 = \omega/c$ is the wave vector in a vacuum with the angular frequency $\omega$ as well as the light speed $c$, and $\theta$ is the incident angle. In Fig. 2a, we sketch a 1D PC (AB)$_N$ composed of the HMM A with substructure dielectric/graphene/metal stacks (CGD)$_S$, and the isotropic dielectric B. Here BeO ($\varepsilon_{BeO} \approx 1.708$)[39] and GaAs ($\varepsilon_{GaAs} \approx 3.48$)[40] are respectively selected for the dielectric layers B and C with slight dispersion near the working frequency (see Supplementary Information, Sec. IX for more details). The indium tin oxide (ITO) is selected for the metal layer D, and its permittivity can be described by the Drude model[41]: $\varepsilon_D = \varepsilon_\infty - \omega_{pD}^2/(\omega^2 + i\omega\gamma_D)$, where $\varepsilon_\infty = 3.9$ is the high-frequency permittivity. Assuming $\hbar\omega_{pD} = 2.48$ eV, $\hbar\gamma_D = 0$ eV in the lossless case and $\hbar\gamma_D = 0.1\hbar\gamma_0 = 0.0016$ eV in the lossy case, $\omega_{pD}$ ($\gamma_D$) denotes the plasma (damping) frequency. Moreover, each layer is presumed as nonmagnetic, i.e. $\mu_B = \mu_C = \mu_D = 1$. Here graphene is selected for the conductive sheet G to provide a modulation DOF involving $\Delta$ and $q_1$, and its surface conductivity can be described as[42–44]: $\sigma_G = i\frac{e^2 E_F}{\pi\hbar^2(\omega + i\tau^{-1})}$, where $\tau = \mu E_F/(e v_f^2)$ is the relaxation rate, and $E_F = \hbar v_f \sqrt{\pi|n|}$ is the Fermi energy (FE). Here $\mu = 10^4$ $cm^2 \cdot V^{-1} \cdot s^{-1}$, $v_f \approx 10^6$ m/s (Supplementary Information, Sec. IX), and $E_F$ can be modulated flexibly through electrostatic doping with tuning charge-carrier density $n$[43].

According to the effective medium theory, the permittivity tensor components of the layer A are given by (Supplementary Information, Sec. I):

$$\varepsilon_{//} = \varepsilon_{Ax} = \varepsilon_{Ay} = \varepsilon_C \delta_C + \varepsilon_D \delta_D + \frac{i\sigma_G}{\varepsilon_0 \omega d}, \varepsilon_\perp = \varepsilon_{Az} = 1/(\delta_C/\varepsilon_C + \delta_D/\varepsilon_D), \quad (1)$$

where $\varepsilon_0$ denotes the vacuum permittivity, $d = d_C + d_D$ denotes the thickness of the unit cell, and $\delta_{C,D} = d_{C,D}/d$ is the filling ratio of the bulk layer C/D. Incidentally, the thickness of each layer is chosen as $d_A = 1210$ nm, $d_B = 2696$ nm, $d_C = 44$ nm and $d_D = 11$ nm, namely $S = 22$ is the number of periods within the HMM. As is shown in Fig. 2b, the structure $(CD)_S$ can be equivalent to an HMM from 230 to 300 THz ($\varepsilon_\parallel / \varepsilon_\perp < 0$), and the real part of $\varepsilon_\parallel$ raises with increasing of the FE from 0.93 to 1.43 eV. We first consider $E_F = 0.93$ eV, and calculate the transmittance spectra of actual structure $[(CGD)_S B]_N$ with the transfer matrix method. In Fig. 2c, d, band degeneracies exist at the point ($f_0 = 288.2$ THz, $k_{\rho 0}/k_0 = 0.7325$) for both polarizations, corresponding to the condition[5]:

$$\alpha^\iota = \frac{\tilde{n}_A^\iota d_A}{\tilde{n}_B^\iota d_B} = \frac{m^\iota}{n^\iota} \in \mathbb{Q} \tag{2}$$

where the superscript $\iota$ represents the transverse-electric (TE) or transverse-magnetic (TM) polarization (playing the role of pseudospins), and $m^{TE} = 7$, $m^{TM} = n^{TE} = n^{TM} = 8$. Here, $\tilde{n}_i^{TE} = \sqrt{\varepsilon_{iy}\mu_{ix} - \frac{\mu_{ix}k_\rho^2}{\mu_{iz}k_0^2}}$ and $\tilde{n}_i^{TM} = \sqrt{\varepsilon_{ix}\mu_{iy} - \frac{\varepsilon_{ix}k_\rho^2}{\varepsilon_{iz}k_0^2}}$ ($i = A, B$) are the effective refractive indexes for TE and TM modes, respectively. The anisotropy between in-plane and out-of-plane directions breaks the electromagnetic duality in the layer A, i.e. $\frac{\varepsilon_\parallel}{\varepsilon_\perp} \neq \frac{\mu_\parallel}{\mu_\perp}$, which provides a theoretical basis for relatively independent and flexible band modulations for two pseudospins. Here the band edges for TE and TM modes (white dotted lines) correspond to $k_z/k_\Lambda = 0.5$ and $k_z/k_\Lambda = 0$ respectively, which rely on the parity of $m^\iota + n^\iota$ ($\iota = $ TE, TM). For example, $(m^{TE} + n^{TE})$ mod $2 = 1$ implies the band edges of $k_z/k_\Lambda = 0.5$ (Supplementary Information, Sec. II). The interval

along $k_z$ denotes such band structure corresponds to the phase of quasi-DNLSs. Moreover, the tilt angles are also distinct. We notice the thickness ratio meets the phase variation compensation effect for TM waves at the frequency of the degenerate point[41], which unveils the unique competition between the negative group velocity coming from the HMM layer A and the positive group velocity coming from the dielectric layer B:

$$d_B/d_A = -\sqrt{\varepsilon_\parallel \varepsilon_B / \mu_\parallel} / \varepsilon_\perp. \tag{3}$$

Thus, the crossing band for the TM (TE) mode corresponds to a unique Type-I (Type-II) WNLS. Then we change the FE of graphene and retrieve the degenerate point based on Eq. (1). In Fig. 2g, the degenerate points for two modes almost move in the same trajectory but with different velocities attributing to the break of duality. From Fig. 2e, f, two WNLSs with different polarizations are separated in the space $E - k_\rho - k_z$, corresponds to the phase of isolated WNLSs, but the topological-symmetry-protected band degeneracies remain stable. In that case, the layers of graphene introduce the DOF of translation into the Weyl quasiparticles.

The break of duality is also manifested in the DOF of rotation. Linked with a fixed point ($f_0^\iota, k_{\rho 0}^\iota$), this parameter group of thicknesses can be denoted as ($d_A, d_B$), and each element in the set $\{(P^\iota d_A/m^\iota, Q^\iota d_B/n^\iota)|\{P^\iota, Q^\iota\} \in \mathbb{Z}^+\}$ will also meet the rational condition in Eq. (2) simultaneously, which turns into $\alpha_{new}^\iota = \frac{P^\iota}{Q^\iota} \in \mathbb{Q}$ naturally. For TM modes, the electric HMM for the layer A provides a negative group velocity, while the dielectric for the layer B provides a positive group velocity. With the change of the ratio of the thicknesses, the group velocity of the whole 1D PC is able to be modulated almost

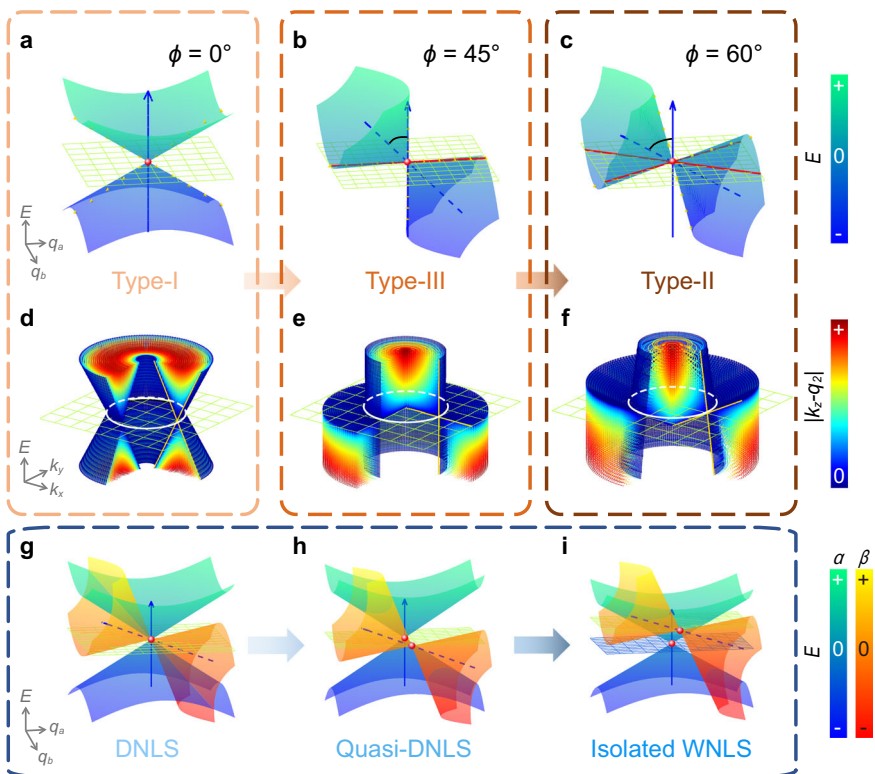

**Fig. 1 | Phase transitions in WNLSs. a–f** illustrate the DOF of rotation. In $E - q_a - q_b$ space, diversified Fermi surfaces (FSs) colored by red: **a** Type-I phase with point-like FS. **b** Critical type-III phase with line-like FS. **c** Type-II phase with cross-like FS. **d–f** The corresponding nodal rings (the white lines) in $E - k_x - k_y$ space evolved from the degenerate points. **g–i** illustrate the DOF of translation. Started from the fourfold DNLS, the Weyl quasiparticle for pseudospin $\beta$ separates from that for pseudospin $\alpha$ in the space of momentum (quasi-DNLS) and energy (isolated WNLSs). The degenerate points and the zero-energy planes are represented by the red points and the mesh surfaces, respectively.

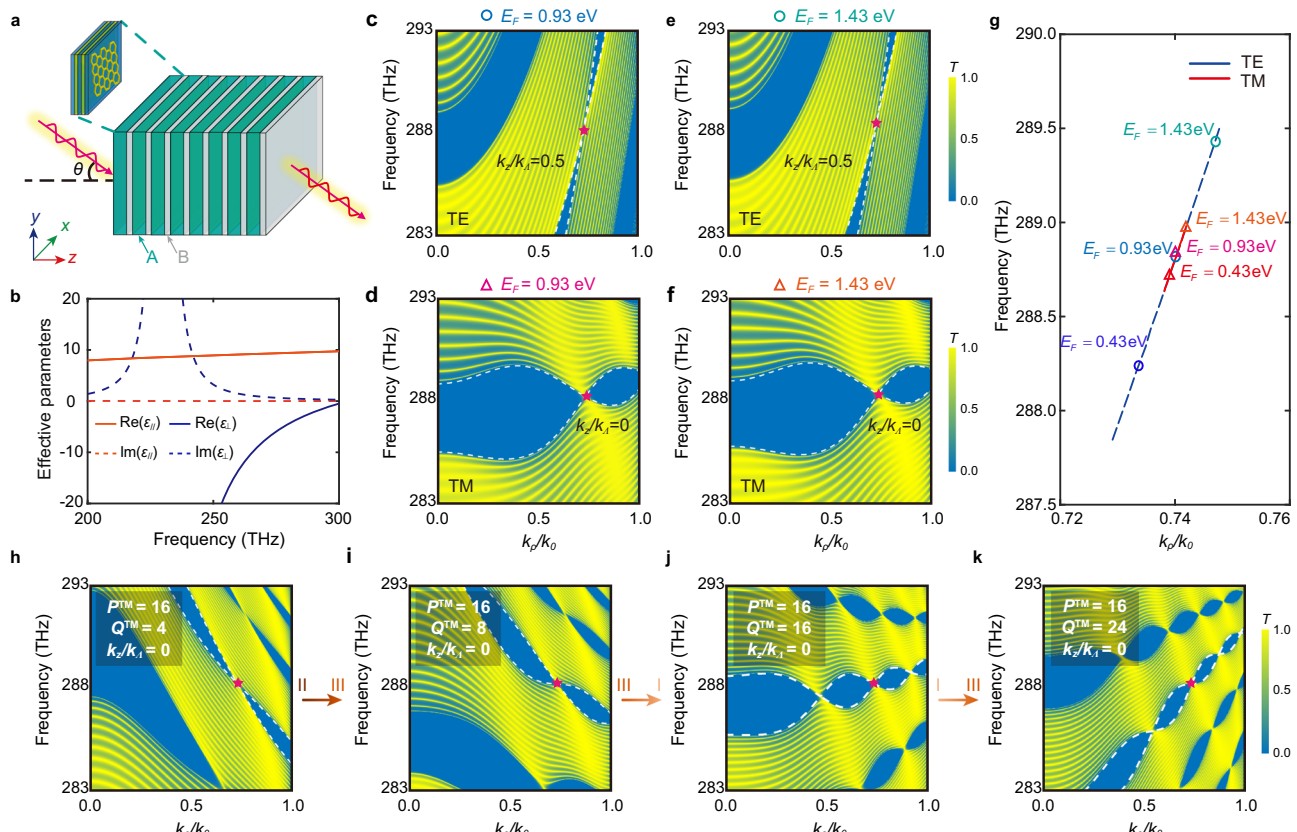

**Fig. 2 | Platform of WNLSs with DOFs of translation and rotation. a** The scheme of a 1D hyper-crystal: $(AB)_N$, where the electric HMM layer A is mimicked by sub-wavelength dielectric/graphene/metal stacks as $(CGD)_S$. **b** Effective permittivity parameters of the structure $(CGD)_S$. **c, d** Transmission spectra of the actual structure $[(CGD)_{22}B]_{20}$ for TE and TM waves with band edges (white dotted lines) $k_z/k_\Lambda = 0.5$ and $k_z/k_\Lambda = 0$ respectively when $E_F = 0.93$ eV (corresponding to the quasi-DNLS) **e, f** Similar to (**c, e**), but $E_F = 1.43$ eV (corresponding to the isolated WNLSs). Here $k_\Lambda = 2\pi/\Lambda$ is the basic reciprocal vectors with the unit cell length $\Lambda = d_A + d_B$. **g** Translation evolution trajectories of the degenerate points with $E_F \in$ [0.1,1.5] eV for TE (blue dashed line) and TM (red solid line) waves. Several momentous points are marked by circles (TE) or triangles (TM) with $E_F = 0.43$ eV, 0.93 eV, 1.43 eV, respectively. For TM waves, the transmittance spectra of the 1D hyper-crystal $[(CGD)_{44}B']_{20}$ with different thickness factors (and $E_F = 0.93$ eV): (**h**) $P^{TM} = 16$, $Q^{TM} = 4$ (type-II WNLS); (**i**) $P^{TM} = 16$, $Q^{TM} = 8$ (type-III WNLS); (**j**) $P^{TM} = 16$, $Q^{TM} = 16$ (type-I WNLS); (**k**) $P^{TM} = 16$, $Q^{TM} = 24$ (type-III WNLS). Here $d_{B'} = \frac{Q^{TM}}{n^{TM}} d_B$, where $n^{TM} = 8$ and $d_B = 2696$ nm. The arrows show the rotation transitions between three types of WNLSs.

continuously within a scope, which induces the unique phase transitions among Type-I/II/III WNLSs, as shown in Fig. 2h−k. Here $d_{A'} = \frac{P^{TM}}{m^{TM}} d_A$ and $d_{B'} = \frac{Q^{TM}}{n^{TM}} d_B$. Noteworthy, taking the effectiveness of Eq. (1) into account, $d_{A'}$ here is regulated by changing the period number $S$ of the HMM layer A, that is $S' = 2S = 44$. We can also get the critical condition of Type-III WNLSs (Supplementary Information, Sec. III):

$$\left(\frac{P^{TM}}{m^{TM}}\right)^2 + \left(\frac{Q^{TM}}{n^{TM}}\right)^2 - \left(\frac{\eta_{Az}^{TM}}{\eta_{Bz}^{TM}} + \frac{\eta_{Bz}^{TM}}{\eta_{Az}^{TM}}\right)\frac{P^{TM}}{m^{TM}}\frac{Q^{TM}}{n^{TM}} = 0, \quad (4)$$

where $\eta_{iz}^{TM} = \frac{n_i^{TM}}{\varepsilon_{ix}}$ ($i$ = A, B) are the impedances for the TM mode. Take the actual structure in Fig. 2 as an example. Since $\frac{\eta_{Az}^{TM}}{\eta_{Bz}^{TM}} + \frac{\eta_{Bz}^{TM}}{\eta_{Az}^{TM}} = 2.2267$ near the degenerate point, we can get two solutions of Eq. (4), $\frac{Q^{TM}}{n^{TM}} = 0.68 \frac{P^{TM}}{m^{TM}}$ together with $\frac{Q^{TM}}{n^{TM}} = 1.4706 \frac{P^{TM}}{m^{TM}}$. Compared with the actual result $\frac{Q^{TM}}{n^{TM}} = 0.5 \frac{P^{TM}}{m^{TM}}$ in Fig. 2i together with $\frac{Q^{TM}}{n^{TM}} = 1.5 \frac{P^{TM}}{m^{TM}}$ in Fig. 2k, the difference may come from the slight dispersions of the layers. Meanwhile, the tilt angles almost remain unchanged with the change of the thickness ratio due to the positive group velocity of both the layer A and B for the TE mode, which are analogous to all-dielectric PCs (Supplementary Information, Sec. III). In that case, the layers of HMMs introduce the DOF of rotation into the Weyl quasiparticles.

Inspired by Feynman's classical thought about nanoparticles, quasiparticles in the form of nodal rings, the simplest closed

topological configurations, may play the role of basic elements and provide a bottom-up method to construct more complicated band structures, like nodal chains, nodal links, and even new topological phases. However, coupling between quasiparticles tends to change their band structure destructively in the process of assembling. To overcome this point, decoupled multi-quasiparticles can provide a flexible platform. On the other hand, coupling[18] like gyrotropic or chiral effects[45] can be introduced actively, which may lead to more DOFs of modulation.

## Bilateral DSS and singularities

From Fig. 2, it can be seen that the phases of double WNLSs are inextricably linked with the positions of degenerate points, and able to be modulated flexibly through the parameters, such as the FE of graphene and the thicknesses of component layers. Then we will discuss the properties of the bandgap detailly. Typically speaking, single-negative (SNG) materials, including epsilon-negative (ENG) and mu-negative (MNG) materials, only support evanescent waves and correspond to the bandgaps. However, a paired structure composed of both ENG and MNG metamaterials will support tunneling mode, namely edge states formed at the boundary between ENG and MNG. For unit cells without inversion symmetry like the structure in Fig. 2a, their bandgaps can carry components of both ENG and MNG simultaneously. To demonstrate this, we illustrate the reflection phases $\varphi_r$ for the TE polarization. As is shown in Fig. 3a, there exist two kinds of photonic insulators at

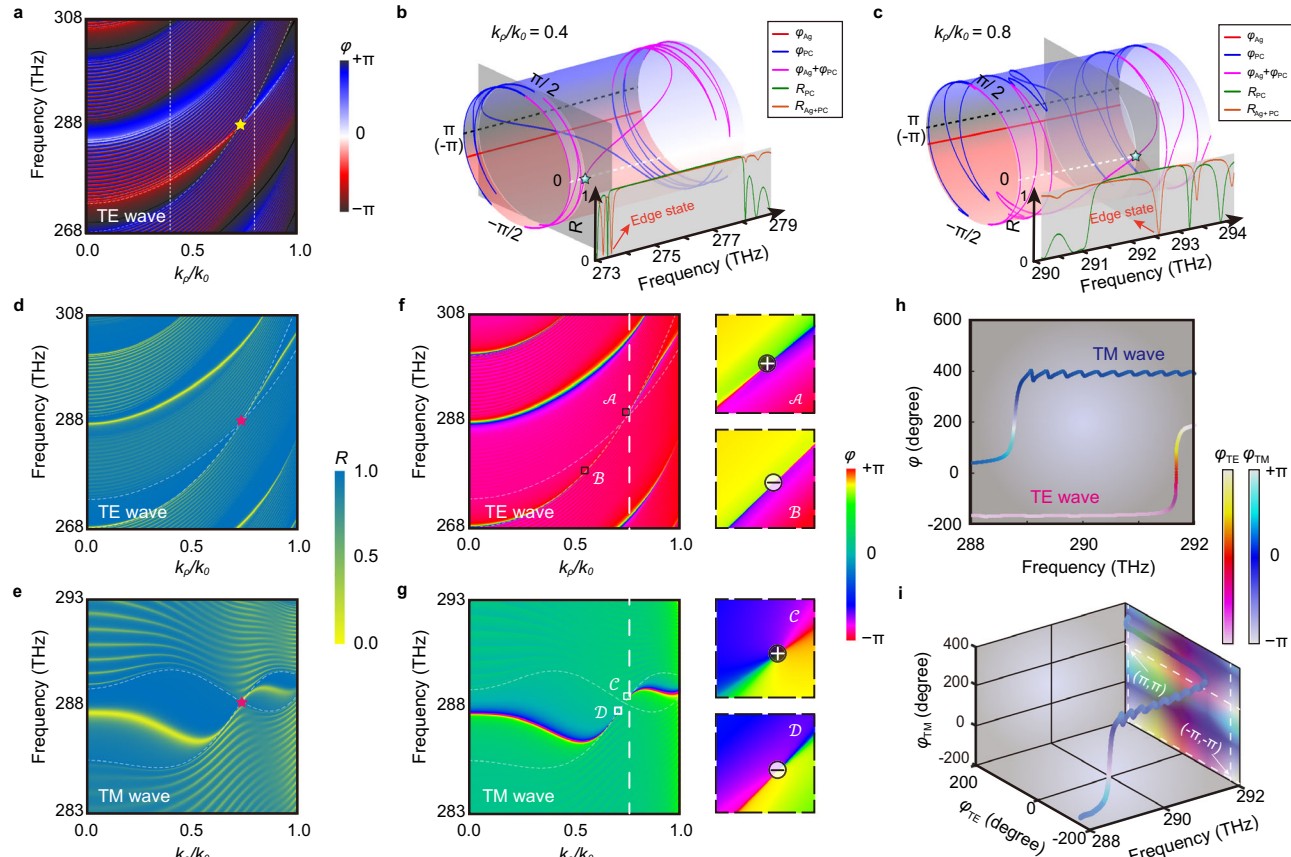

**Fig. 3 | Bilateral DSSs in double WNRs. a** Phase diagrams $\varphi_r$ of the PC [(CGD)$_{22}$B]$_{20}$ for the TE wave. **b, c** When $k_\rho/k_0 = 0.4$ and $k_\rho/k_0 = 0.8$, the reflection phases $\varphi_{Ag}$ for the silver layer E (red), $\varphi_{PC}$ for the PC (blue) together with their sum $\varphi_{Ag} + \varphi_{PC}$ (magenta) are illustrated on the cylinders, and related reflections $R_{PC}$ (green) together with $R_{Ag+PC}$ (brown) are illustrated in the insets. The boundary of $\varphi_r = \pi(-\pi)$ and the center of $\varphi_r = 0$ are highlighted by black and white dotted lines, respectively. The points satisfied $\varphi_{Ag} + \varphi_{PC} = 0$ are marked by cyan stars.

**d, e** Reflection spectra and **f, g** phase diagrams $\varphi_r$ of the hyper-crystal E[(CGD)$_{22}$B]$_{20}$ for TE and TM waves. The surface states support four singularities near the degenerate points: $\mathcal{A}$ and $\mathcal{C}$ with +1 topological charge, together with $\mathcal{B}$ and $\mathcal{D}$ with −1 topological charge. **h** For $\theta = 52^o$, phase jumps for TE and TM waves emerge respectively near 291.6 THz and 288.8 THz, respectively. **i** Phase variations illustrated by a joint 2D colormap in an expansion space $\varphi_r^{TE} - \varphi_r^{TM}$.

the same bandgap, respectively corresponding to ENG $\varphi_r \in [-\pi, 0]$ (gradient red) and MNG $\varphi_r \in [0, \pi]$ (gradient blue) phases, which are separated by black lines ($\varphi_r = \pm\pi$)[46]. Then, we add an extra 20 nm silver layer E before the incident interface of PC in Fig. 2a, and its permittivity can be described by the Drude model $\varepsilon_E = \varepsilon_\infty - \omega_{pE}^2/(\omega^2 + i\omega\gamma_E)$ where $\varepsilon_\infty = 4.09$, $\omega_{pE} = 1.33 \times 10^{16}$ rad/s, and $\gamma_E = 1.33 \times 10^{14}$ rad/s. As an example, the reflection phases of $k_\rho/k_0 = 0.4$ and $k_\rho/k_0 = 0.8$ are plotted respectively in Fig. 3b, c, belonging to the regions inside and outside the nodal ring. When we cut a period $[-\pi, \pi]$ of the axis $\varphi_r$, twist and glue end to end, the plane of $f - \varphi_r$ is molded into a cylinder, where the upper region (gradient blue) and the lower region (gradient red) correspond to MNG and ENG phases, respectively. There is no doubt the metal layer E (the red line) corresponds to the ENG phase. And for $k_\rho/k_0 = 0.4$, the condition of a stable DSS $\varphi_{Ag} + \varphi_{PC} = 0$[27,47] is satisfied near $f = 273.52$ THz (the cyan star), corresponding to a dip of the reflection $R_{Ag+PC}$ (the orange line) in Fig. 3b. Here $\varphi_{Ag}$ and $\varphi_{PC}$ are the reflection phases $\varphi_r$ of the silver layer E and the PC [(CGD)$_{22}$B]$_{20}$, respectively, $\varphi_{Ag} + \varphi_{PC}$ is their sum, and $R_{Ag+PC}$ corresponds to the reflection of the composite structure E[(CGD)$_{22}$B]$_{20}$. Similarly, for $k_\rho/k_0 = 0.8$, the corresponding frequency changes into $f = 292.44$ THz. Analogous conclusions can be obtained for the TM polarization. Nevertheless, $\varphi_r \in [-\pi, 0]$ ($\varphi_r \in [0, \pi]$) corresponds to the MNG (ENG) phase currently (Fig. S5 in Supplementary Information, Section III). In that case, the DSSs exist both outside and inside the nodal ring, as shown in Fig. 3d, e. (More details about

unilateral DSSs when restoring the inversion symmetry can be seen in Fig. S6 in Supplementary Information, Section IV) Their topological properties can also be characterized by the reflection phase $\varphi_r$. In Fig. 3f, two singularities for the TE wave exist near the points $\mathcal{A}$ ($f_0 = 291.47$ THz, $k_{\rho0}/k_0 = 0.78$) and $\mathcal{B}$ ($f_0 = 277.75$ THz, $k_{\rho0}/k_0 = 0.53$), which are divided from an ideal BIC with trivial topological charge (Supplementary Information, Sec. V). By tracing an anticlockwise closed loop around these points, two singularities $\mathcal{A}$ and $\mathcal{B}$ carry integer topological charges characterized by winding number $v = (1/2\pi) \oint d\varphi_r = +1$ and $-1$ respectively, which means the reflection phase $\varphi_r$ can precisely 'wind' around the cylinder for one time (Fig. S7 in Supplementary Information, Sec. V). Since the starting and terminal points are coincident, the permitted winding numbers are quantized on the cylinder, while the winding direction determines the sign of topological charges. Similarly, +1 and −1 charges emerge near the points $\mathcal{C}$ ($f_0 = 288.51$ THz, $k_{\rho0}/k_0 = 0.75$) and $\mathcal{D}$ ($f_0 = 287.62$ THz, $k_{\rho0}/k_0 = 0.69$) for the TM wave, as shown in Fig. 3g. Noteworthy, there are two different properties discussed within the framework of reflection phases: quantity of $\varphi_r$ with blue-red color bar in Fig. 3a illustrates the classification of bandgaps (ENG or MNG), while variation of $\varphi_r$ with rainbow color bar in Fig. 3f, g illustrates the existence of singularities. Near the singularity, the phase changes dramatically. At $\theta = 52^o$ ($k_{\rho0}/k_0 = 0.79$), Heaviside-like phase jumps occur near 291.6 THz for the TE modes and 288.8 THz for the TM modes, as is shown in Fig. 3h. In fact, such jumps are quite sensitive to environmental

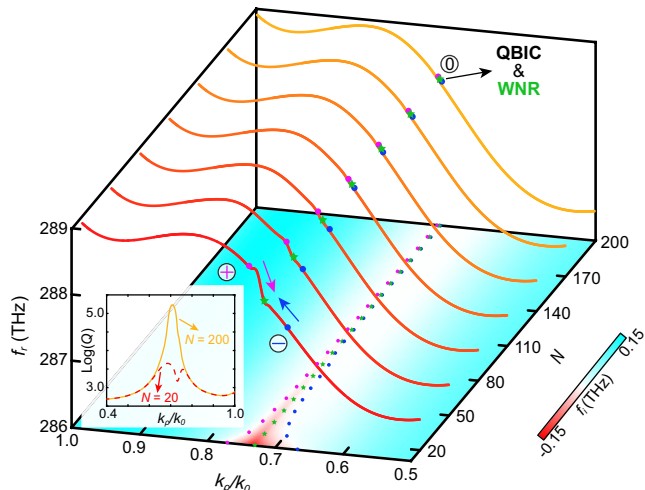

**Fig. 4 | Degeneracy features of WNRs and singularities.** Reflection-zero dispersion for bilateral DSS of the PC: $[(CGD)_{22}B]_N$ in the complex frequency space $f = f_r + if_i$ for the TM wave. The positions of reflection-phase singularities with positive (negative) topological charges are indicated by purple (blue) circles, while those of WNRs are indicated by green stars. The inset: the quality factors $Q$ with the unit cell number of $N = 20$ (the red dashed line) and $N = 200$ (the yellow solid line).

disturbances, such as the external refractive indexes[33,48], molecule attachments[49,50], and temperature[51], which lays a solid foundation for sensors with ultrahigh sensitivities. However, the presence of singularities mainly depends on the TM component (zero points of the complex reflection ratio $\rho = r^{TM}/r^{TE}$) in most of the previous schemes, while the TE component contribution (pole points) is negligible. Inheriting the decoupled properties from multi-quasiparticles, different evolutions for two polarizations determine that we can review the sensing scheme in expansion space $\varphi_r^{TE} - \varphi_r^{TM}$. Here we employ a joint 2D colormap to reveal the properties of this dual mode in Fig. 3i, which are expected to support a highly sensitive interferometric sensing scheme with a unique dual mode (Supplementary Information, Section VII).

On the other hand, it can be predicted that when two singularities with different topological charges merge in the momentum space, the total charge will become zero due to charge conservation. Here such a phenomenon is observed in the space of complex frequency $f = f_r + if_i$. Out of completeness, we take the TM polarization as an example. Fig. 4 shows the complex frequency as the solution of the reflection-zeros, and $f_r$ ($f_i$) represents the real (imaginary) part of the solution. For $N = 20$, the intersections between the dispersion and the axis of $f_i = 0$, which symbolize pure real excitation frequencies of the surface states, are marked by purple and blue circles near the points $k_\rho/k_0 = 0.75$ and $k_\rho/k_0 = 0.69$, corresponding to the singularities C and D in Fig. 3g. When increasing the number of the unit cell ($N = 200$), the leakages of radiation reduce, and the singularity pair merges near the degenerate point $k_\rho/k_0 = 0.73$ of the WNR (marked by green stars), which forms the quasi-BIC. Besides, the quality factor $Q = f_0/\Delta f$ is another important evidence to judge singularities, where $f_0$ is the resonant frequency and $\Delta f$ is the full width at half maximum (FWHM). Exactly, peaks of the quality factor appear near the position of singularities as shown in the inset of Fig. 4, which implies most energy is localized in the composite structure. Analogous conclusions can be obtained for the TE mode at the same point. Each above singularity pair comes from the same nodal line (marked by green stars) corresponding to the ultimate quasi-BIC. Similar situations appear in the structures of isolated WNLSs and DNLSs, respectively, which may give rise to a flexibly controllable BIC (Supplementary Information, Section V). Therefore, DSSs become an intriguing bridge between Weyl semimetals and BICs. Worth

mentioning, we notice a recent work discusses BICs spawned from the Dirac point in the PC slab[52]. They find "the eigenstates can be mixed to any ratio to produce any amplitudes of diffraction", including the BIC, which may provide a different perspective to the BICs originated from decoupled multi-quasiparticles, and reveal a universal relevance between the nodal physics and the singularity physics.

## Discussion

As a summary, we establish a 1D PC platform to realize manipulations of one kind of Weyl quasiparticles (double WNRs) with the properties of BICs both dynamically and topologically. Based on four DOFs, phase transitions of translation (from isolated WNLSs to quasi-DNLSs and DNLSs) and rotation (from type-I to critical type-III and type-II WNLSs) are realized by flexibly modulating the FE of graphene and the thicknesses of component layers, respectively. In particular, when such a structure is truncated by a metal film, extra DSSs pinned by the nodal lines can support degenerate quasi-BICs with two pseudospins. Tuning the absorption losses and radiation leakages, each BIC can divide into two reflection-phase singularities with opposite topological charges. With flexible and stable phase jumps, the approaching singularities supported by degenerate BICs may improve the traditional sensing schemes. This work opens an unexplored avenue to bridging BICs and WNRs via hyper-crystal, giving a promising way for applications on topological photonics.

## Methods
### Hamiltonian models

To manipulate the Weyl quasiparticles in the energy-momentum space flexibly, we introduce four modulation DOFs $\Delta$, $\phi$, $q_1$ and $q_2$ into the general model of a single WNR $H = (q_a - q_1)\sigma_z + (q_b - q_2)\sigma_x$. For Fig. 1a–f, the effective Hamiltonian can be given by

$$H = \Delta\sigma_0 + (q_a - q_1)\xi + \frac{q_b - q_2}{\sqrt{\cos(2\phi)}}\sigma_x, \tag{5}$$

where $\xi = \tan(2\phi)\sigma_0 + \sec(2\phi)\sigma_z$. $\sigma_i(i = 0, x, y, z)$ denotes the Pauli matrix. Furthermore, we can take the form of direct sum to describe decoupled multi-quasiparticles. Take double decoupled WNRs for different pseudospins $\alpha$ and $\beta$ as an example, the general Hamiltonian, corresponding to Fig. 1g–i, can be given by

$$H_{II} = H_I^\alpha(\Delta^\alpha, \phi^\alpha, q_1^\alpha, q_2^\alpha) \oplus H_I^\beta(\Delta^\beta, \phi^\beta, q_1^\beta, q_2^\beta) = \tau_0 \otimes \{X + Y\} + \tau_z \otimes \{M + N\}, \tag{6}$$

where $X = a^+\sigma_0 + b^+\sigma_z$, $Y = c^+\sigma_x$, $M = a^-\sigma_0 + b^-\sigma_z$, and $N = c^-\sigma_x$. In addition, $a^\pm = [\Delta^\alpha \pm \Delta^\beta + \delta q_\alpha \tan(2\phi^\alpha) \pm \delta q_\beta \tan(2\phi^\beta)]/2$, $b^\pm = [\delta q_\alpha \sec(2\phi^\alpha) \pm \delta q_\beta \sec(2\phi^\beta)]/2$, and $c^\pm = \frac{q_b - q_2^\alpha}{2\sqrt{\cos(2\phi^\alpha)}} \pm \frac{q_b - q_2^\beta}{2\sqrt{\cos(2\phi^\beta)}}$. Note that $\tau_i$ and $\sigma_i$ ($i = 0, x, y, z$) represent the Pauli matrix for the band index and pseudospin index, respectively.

## Data availability

The data generated in this study have been deposited in Figshare database under the following accession code https://doi.org/10.6084/m9.figshare.25294858.

## Code availability

The code that supports the plots within this paper can be found in Figshare database under the following accession code https://doi.org/10.6084/m9.figshare.25294858.

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

## Acknowledgements

This work is supported by the National Key R&D Program of China (Nos. 2023YFA1407600 and 2021YFA1400602), the National Natural Science Foundation of China (Nos. 12004284 and 12374294), the Fundamental Research Funds for the Central Universities (No. 22120210579), and the Chenguang Program of Shanghai (No. 21CGA22).

## Author contributions

Z. Guo conceived the idea. S. Hu and Z. Guo proposed the model, performed the numerical simulations and theoretical analyses. Z. Guo, S. Chen, and H. Chen supervised the whole project. S. Hu, Z. Guo, and W. Liu wrote the manuscript. All authors contributed to discussions of the results and the manuscript.

## Competing interests

The authors declare no competing interests.
