## [Peer Review File · Nature Communications]

REVIEWER COMMENTS

Reviewer #2 (Remarks to the Author):

In this work, Hu et al. propose a photonic double Weyl nodal line semimetals with an effective Hamiltonian determined by four degrees of freedom in one-dimensional hyper-crystals. They first demonstrate a double Weyl nodal ring structure by simulating energy band structures of TE and TM mode, and then they calculate the reflection phase diagram, resulting in a bilateral drumhead surface state and a pair of reflection-phase singularities near each nodal line. Furthermore, after reducing radiation leakages and absorption losses, they illustrate that two singularities gather together gradually, and claim that these singularities form a bound state in the continuum ring at the nodal line ultimately. The manuscript is well-written, but I still have a few questions on the content of this work.

In Fig. 3 (b) (c), why is it necessary to fold the $f\text{-}\phi_r$ diagram into a cylinder-shaped diagram? Readers may find the diagram hard to read if it is cylinder-shaped. Also, the paper doesn't explain the physical meaning of ϕ_{Ag} , ϕ_{PC} and $\phi_{Ag+\phi_{PC}}$, while the simulated data of ϕ_{Ag} and ϕ_{PC} seem to be unimportant and unnecessary to show out, since only $\phi_{Ag+\phi_{PC}}$ determines the edge state. Furthermore, Figs. 3 (a) (f) (g) illustrate the same physical parameter, but they have different representation in the color bars; (f) (g) illustrate the same system in different modes, but they have different scaling in the frequency axis. These differences will possibly cause readers to feel confused and hard to follow.

From Line 117 to Line 124, the paper starts to discuss the concept of "decoupled multi-quasiparticles," which has later been further emphasized through the proposed double Weyl nodal ring structure in TE and TM modes. However, although the construction of the topological structure has been successful, the importance of this topic has not been sufficiently discussed. Is there any further photonic application or physical implication of this concept in other systems? The paper eventually reached a BIC ring by reducing radiation leakages and absorption losses in its final part. Does the concept of "decoupled multi-quasiparticles" relate to the construction of the finally achieved BIC ring?

In Line 265, the paper claims that the singularity pairs near the nodal line will eventually form BIC when reducing radiation leakages and absorption losses. However, is there any theoretical or calculational proof that the BIC is actually formed in this situation? Readers may not be fully convinced that there actually exists a BIC even when the singularities become very close to each other.

Reviewer #3 (Remarks to the Author):

In the manuscript entitled "Hyperbolic Metamaterial Empowered Controllable Photonic Weyl Nodal Line Semimetals", Hu et al. reported the realization of quasi-DNLS, isolated WNLS, etc. by using a 1D hyper-crystal containing hyperbolic metamaterials. Especially, for the TM modes, phase transitions between three types of WNLS can be achieved through the change of the ratio between the thickness of layer A and layer B. Then, by introducing a silver layer at the surface, the structure can support drumhead surface state (DSS) with phase singularities for both TE and TM polarizations. The generated phase singularities will approach each other and eventually merge with the increase of the number of the unit

cell.

This manuscript presents good theoretical analyzations and convincing numerical simulations. I believe that this idea proposed by the authors is interesting and provides a new approach to realize varieties of topological effects in photonic systems. I would like to recommend the publication of this manuscript in Nature Communications.

Some minor points:

1. Is there any difference of the phase singularities in Figure 3 for different thicknesses of the sliver layer E?

2. It's interesting that the phase singularities will evolve with the increasing number of the unit cell, and the quality factor will also increase to an ultrahigh value. For traditional BICs, they usually correspond to modes with infinite quality factor. Here, I may suggest the high-Q modes in this manuscript to be called quasi-BICs.

Response Letter to Reviewers

We are grateful for the useful comments on this manuscript (ID: NCOMMS-23-24067) from all the reviewers. Based on the reviewers' instructive suggestions, we revise our paper and address all the points raised by the reviewers as listed below. In the text below each of the comments from each reviewer is quoted and is followed by the corresponding detailed response. We also revised the manuscript and the Supplementary Information accordingly, and these updates are highlighted in those files.

Main Change List

1. Replot Fig. S7 as direct comparisons between cylinder-shaped and corresponding planar diagrams. The relevance between the cylinder-shaped diagrams and the topological invariants (winding number) characterizing the singularities is stressed in the main text.
2. Replot Figs. 3(b) and 3(c) to make the cylinder-shaped diagrams clearer.
3. Clarify the physical meaning of φ_{Ag} , φ_{PC} , and $\varphi_{Ag} + \varphi_{PC}$ in the main text.
4. Add Fig. S7 to the Supplementary Materials to further reveal the relevance between the distribution of φ_{PC} , the inversion symmetry, and the existence of the edge states.
5. Clarify the difference between two color bars in Figs. 3(a) and 3(f)–3(g) and their corresponding usage scenarios.
6. Add a new section to envisage the further photonic applications and physical implications of the concept of “decoupled multi-quasiparticles” into the main text.
7. Add the calculation results to demonstrate that the BIC tends to be formed in the nodal line when singularities with opposite topological charges come close to each other.
8. Add Fig. S16 to the Supplementary Materials to depict the evolution and even annihilation of singularities through altering the thickness of the silver layer E.
9. Revise the description of high-Q modes with “quasi-BICs”.

Reviewer #2 (Remarks to the Author):

In this work, Hu et al. propose a photonic double Weyl nodal line semimetals with an effective Hamiltonian determined by four degrees of freedom in one-dimensional hyper-crystals. They first demonstrate a double Weyl nodal ring structure by simulating energy band structures of TE and TM mode, and then they calculate the reflection phase diagram, resulting in a bilateral drumhead surface state and a pair of reflection-phase singularities near each nodal line. Furthermore, after reducing radiation leakages and absorption losses, they illustrate that two singularities gather together gradually, and claim that these singularities form a bound state in the continuum ring at the nodal line ultimately.

Response: We thank the reviewer for his/her careful reading and nice summary of our manuscript. We also thank this reviewer for raising the following important comments, and for indicating explicitly the ways to improve the presentation of this manuscript.

The manuscript is well-written, but I still have a few questions on the content of this work.

Response: We thank the reviewer for his/her recognitions and positive assessments of this manuscript. In particular, the reviewer considers that "The manuscript is well-written". Based on the reviewer's excellent suggestions, we have carefully addressed all the comments in the revised paper. The following is a point-to-point response to the reviewer's comments.

1. In Fig. 3 (b) (c), why is it necessary to fold the $f-\varphi_r$ diagram into a cylinder-shaped diagram? Readers may find the diagram hard to read if it is cylinder-shaped.

Response: Thanks for the reviewer’s concerns. This is a crucial point and we are happy to clarify it. In fact, although the cylinder-shaped diagram can be mapped to a plane diagram here, the presentation of a cylinder-shaped diagram is beneficial for observing the topological order of singularities and the single-negative characteristics of bandgaps:

(1) The topological invariant defined by $v = (1/2\pi) \oint d\varphi_r$ is winding number, which characterizes singularities (topological charges) discussed in Figs. 3(c) and 3(d). Take the +1 singularity \mathcal{A} for the TE wave as an example, we notice tracing an anticlockwise loop $\mathcal{A}_1\mathcal{A}_2\mathcal{A}_3\mathcal{A}_4\mathcal{A}_1$ enclosing the singularity \mathcal{A} [Fig. R1(a)], the reflection phase φ_r of the composite structure $E[(CGD)_{22}B]_{20}$ could precisely “wind” around the cylinder for one time [Fig. R1(b)]. Similar phenomena could be found near the -1 singularity \mathcal{B} [Fig. R1(d)], but its winding direction is just opposite to that of \mathcal{A} [Fig. R1(e)]. Therefore, this cylinder-shaped diagram provides a geometrical presentation to capture the concept of winding number.

(2) Considering the plane diagram, the range of reflection phase φ_r tends to beyond $[-\pi, \pi]$ when ensuring the continuity of the lines, as is shown in Figs. R1(c) and R1(f). Hence, we have to introduce an extra phase $2n\pi$ ($n \in \mathbb{Z}$) into the conditions for a planar diagram, like epsilon-negative (ENG) materials with $\varphi_r \in [(2n - 1)\pi, 2n\pi]$ and mu-negative (MNG) materials with $\varphi_r \in [2n\pi, (2n + 1)\pi]$ for the TE mode. Mapping $f - \varphi_r$ diagram to a cylinder could not only exhibit the single-negative characteristics of bandgaps, but also avoid redundant descriptions about the periodicity of φ_r to some extent.

In order to better guide the readers, we have added the following discussion (in Line 248) into the revised manuscript to further stress the relevance between the cylinder-shaped diagram and the winding number, and supplemented Figs. R1(b)–R1(c) together with R1(e)–R1(f) in the supplementary information, Sec. V (in Page 18) as direct comparisons between cylinder-shaped and corresponding planar diagrams.

“By tracing an anticlockwise closed loop around these points, two singularities \mathcal{A} and \mathcal{B} carry integer topological charges characterized by winding number $v = (1/2\pi) \oint d\varphi_r = +1$ and -1 respectively, which means the reflection phase φ_r can precisely ‘wind’ around the cylinder for one time (Fig. S7 in Supplementary Information, Sec. V). Since the starting and terminal points are coincident, the permitted winding numbers are quantized on the cylinder, while the winding direction determines the sign of topological charges.”

Figure R1 (FIG. S7 in the supplementary information). Phase diagrams φ_r of the composite structure $E[(CGD)_{22}B]_{20}$ for the TE wave. (a) The region near the singularity \mathcal{A} is surrounded by an anticlockwise rectangle closed loop $\mathcal{A}_1\mathcal{A}_2\mathcal{A}_3\mathcal{A}_4\mathcal{A}_1$. (b) Reflection phase variation with topological charge (winding number) +1 when tracing the loop on the cylinder, and (c) similar variation with more fixed incident angles near the singularities \mathcal{A} . (d)–(f) Similar to (a)–(c), but around the singularity \mathcal{B} with topological charge (winding number) -1.

Moreover, we have made efforts to add a circular axis of φ_r on the left, and adjust the perspective of cylinder-shaped diagrams to make the cylinder-shaped diagram clearer, as shown in Fig. R2.

Figure R2 (Figures 3b and 3c in the revised main text). Bilateral drumhead surface states (DSSs) in double WNRs. (a) Phase diagrams φ_r of the PC $[(CGD)_{22}B]_{20}$ for the TE wave. When (b) $k_\rho/k_0 = 0.4$ and (c) $k_\rho/k_0 = 0.8$, the reflection phases φ_{Ag} for the silver layer E (red), φ_{PC} for the PC (blue) together with their sum $\varphi_{Ag} + \varphi_{PC}$ (magenta) are illustrated on the cylinders, and related reflections R_{PC} (green) together with R_{Ag+PC} (brown) are illustrated in the insets. The boundary of $\varphi_r = \pi(-\pi)$ and the center of $\varphi_r = 0$ are highlighted by black and white dotted lines, respectively. The points satisfied $\varphi_{Ag} + \varphi_{PC} = 0$ are marked by cyan stars.

2. Also, the paper doesn't explain the physical meaning of φ_{Ag} , φ_{PC} and $\varphi_{Ag} + \varphi_{PC}$, while the simulated data of φ_{Ag} and φ_{PC} seem to be unimportant and unnecessary to show out, since only $\varphi_{Ag} + \varphi_{PC}$ determines the edge state.

Response: Thanks for the reviewer's concerns.

(1) We are very sorry that the physical meaning of φ_{Ag} , φ_{PC} , and $\varphi_{Ag} + \varphi_{PC}$ was not clearly explained in the previous version. In fact, φ_{Ag} and φ_{PC} are the reflection phases φ_r of the silver layer E and the PC $[(CGD)_{22}B]_{20}$, respectively. And $\varphi_{Ag} + \varphi_{PC}$ is their sum. In the revised manuscript, we have added the following sentence into the main text (in Line 236) to make their physical meaning clearer.

“Here φ_{Ag} and φ_{PC} are the reflection phases φ_r of the silver layer E and the PC $[(CGD)_{22}B]_{20}$ respectively, $\varphi_{Ag} + \varphi_{PC}$ is their sum, and R_{Ag+PC} corresponds to the reflection of the composite structure $E[(CGD)_{22}B]_{20}$.”

(2) We fully agree with the reviewer that $\varphi_{Ag} + \varphi_{PC}$ determines the edge state. Nevertheless, we hope to display the reflection phases φ_{Ag} and φ_{PC} as far as possible, because these extra parameters might bring readers a step-by-step picture to follow the formation mechanism of the edge states. On the one hand, we hope the reflection phase φ_{Ag} for the silver (well-known as ENG materials) layer E becomes an intuitive indicator to identify the regions of ENG and MNG on the phase cylinder. In particular, the regions of ENG (MNG) are just contrary for the TE [Figs. R2(b)–R2(c)] and TM [Figs. R3(b)–R3(c)] waves. Especially, ENG with $\varphi_r \in [-\pi, 0]$ and MNG with $\varphi_r \in [0, \pi]$ for the TE mode, while ENG with $\varphi_r \in [0, \pi]$ and MNG with $\varphi_r \in [-\pi, 0]$ for the TM mode.

Figure R3. (FIG. S5 in the supplementary information) (a) Phase diagrams φ_r of the PC $[(CGD)_{22}B]_{20}$ for the TM wave. When (b) $k_\rho/k_0 = 0.4$ and (c) $k_\rho/k_0 = 0.8$, the calculated reflection phase φ_{Ag} (red), φ_{PC} (blue) together with $\varphi_{Ag} + \varphi_{PC}$ (magenta) is illustrated on the cylinder, and related reflection R_{PC} (green) together with R_{Ag+PC} (brown) is illustrated on the vertical rectangular window. On the cylinder, the upper region (colored by gradient blue) corresponds to the ENG phase, while the lower region (colored by gradient red) corresponds to the MNG phase, where the boundary of $\varphi_r = \pi(-\pi)$ and the center of $\varphi_r = 0$ are highlighted by black and white dotted lines, respectively. The points satisfied $\varphi_{Ag} + \varphi_{PC} = 0$ are marked by cyan stars.

On the other hand, we notice φ_{Ag} nearly remains invariable near our interested frequency f and wave vector k_ρ , which means the formation of the edge states (tunneling modes at the boundary between ENG and MNG) are tightly related to φ_{PC} . To elucidate this, we try to restore the inversion symmetry of the PC to change the distribution of φ_{PC} . As is shown in Fig. R4, PC with the structure $(B''AB'')_{20}$ is considered. Compared with the structure $(AB)_{20}$ in Fig. 2(a), $d_{B''} = 0.5d_B$ here, and other parameters remain unchanged. From the diagram of reflection phase φ_r for the PC in Fig. R4(a), each bandgap for the TE wave, inside or outside the Weyl nodal ring (WNR), can only support one component, ENG (gradient red) or MNG (gradient blue) respectively. In that case, the outside bandgap region [Fig. R4(c)] can meet the

condition $\varphi_{Ag} + \varphi_{PC} = 0$, rather than the inside bandgap region [Fig. R4(b)], which leads to a unilateral drumhead surface state (DSS) [Fig. R4(d)]. Similar situations can be found for the TM wave. At this moment, it comes back to the case discussed in Ref. [R1].

In order to better guide the readers, in the revised manuscript, we have added the following discussion into the supplementary information, Sec. IV (in Page 16) to further reveal the relevance between the inversion symmetry, the distribution of φ_{PC} and the existence of the edge states. Therefore, we request the reviewer to allow us to retain the results of φ_{Ag} and φ_{PC} in the manuscript.

“In that case, a pair of edge states is supported inside and outside the Weyl nodal ring (WNR), corresponding to the bilateral drumhead surface state (DSS). As a comparison, we modify the unit cell of PC from the form of (AB) to (B''AB'') to restore the inversion symmetry. Here $d_{B''} = 0.5d_B$, and other parameters remain unchanged with the structure (AB)₂₀ in Fig. 2(a). As is shown in Fig. S6(a), the bandgap region for PC, inside or outside the WNR, is filled with only one component, ENG (gradient red) or MNG (gradient blue), respectively. Hence, the condition $\varphi_{Ag} + \varphi_{PC} = 0$ is only met outside the WNR in Figs. S6(b)–(c), leading to a unilateral DSS ultimately in Fig. S6(d). Similar situations can be found for the TM wave in Figs. S6(e)–(h). At this moment, it comes back to the case discussed in Ref. [S6].”

Figure R4. (FIG. S6 in the supplementary information) Unilateral DSSs in double WNRs. (a) Phase diagram φ_r of the PC $(B''AB'')_{20}$ with inversion symmetry for the TE wave. Compared with the structure $(AB)_{20}$ in Fig. 2(a), $d_{B''} = 0.5d_B$ here, and other parameters remain unchanged. (b) When $k_\rho/k_0 = 0.4$, the reflection phases φ_{Ag} for the silver layer E (red), φ_{PC} for the PC (blue) together with their sum $\varphi_{Ag} + \varphi_{PC}$ (magenta) are illustrated on the cylinders, and related reflections R_{PC} (green) together with R_{Ag+PC} (brown) are illustrated in the insets. The boundary of $\varphi_r = \pi(-\pi)$ and the center of $\varphi_r = 0$ are highlighted by black and white dotted lines, respectively. (c) Similar to (b), but for $k_\rho/k_0 = 0.8$. The unique point satisfied $\varphi_{Ag} + \varphi_{PC} = 0$ is marked by cyan stars. (d) Reflection spectra of the PC $(B''AB'')_{20}$ for the TE wave. The rest is shaded except the interested gap, and the inset is an enlarged view. DSS only exists in the region outside the WNR. (e)–(h) Similar to (a)–(d), but for the TM wave.

3. Furthermore, Figs. 3 (a) (f) (g) illustrate the same physical parameter, but they have different representation in the color bars; (f) (g) illustrate the same system in different modes, but they have different scaling in the frequency axis. These differences will possibly cause readers to feel confused and hard to follow.

Response: Thanks for the reviewer's concerns. These are important points and we are happy to clarify it.

(1) About the **color bars**: In fact, the purpose of using two color codes is to clearly describe two different properties within a unified framework of physical parameter, reflection phase φ_r in Fig. 3. On the one hand, local properties of the photonic crystal depend on the quantity of reflection phase, which could illustrate the classification of bandgaps, as shown in Fig. 3(a). Especially, bandgaps are opaque and equivalent to single-negative (SNG) photonic insulators, including ENG (permittivity $\varepsilon < 0$ and permeability $\mu > 0$) and MNG (permittivity $\varepsilon > 0$ and permeability $\mu < 0$) materials [R2]. They correspond to two different regions on the axis of φ_r : Take TE wave as an example, ENG with $\varphi_r \in [-\pi, 0]$ and MNG with $\varphi_r \in [0, \pi]$. Therefore, the blue-red color bar could provide distinct boundaries to depict this duality. On the other hand, the global and dynamic properties depend on the variation of reflection phase, which could illustrate the existence of singularities shown in Figs. 3(f) and (g). By tracing a closed loop around one singularity, a total accumulated phase of $\pm 2\pi$ could be obtained [R3]. This property could help us to locate the position of singularities, and the rainbow color bar contributes to distinguishing the vortex texture of reflection phases near one singularity.

In the revised manuscript, we have added the following discussion into the main text (in Line 256) to further clarify the difference between two color bars and corresponding usage scenarios.

“Noteworthy, there are two different properties discussed within the framework of reflection phases: quantity of φ_r with blue-red color bar in Fig. 3a illustrates the classification of bandgaps (ENG or MNG), while variation of φ_r with rainbow color bar in Fig. 3f–g illustrates the existence of singularities.”

We also tried to unify the color bars of reflection phase. However, the reflection phase represented by a unified color bar cannot clearly express the two types of physical properties mentioned above. For example, the rainbow color bar, as is shown in Fig. R5(a) [corresponding to Fig. 3(a)], seems too complicated to identify the regions belonging to ENG or MNG, while the blue-red color bar, as is in Fig. R5(b) [corresponding to Fig. 3(f)], seems too simple to capture the dramatical Heaviside-like phase jumps [shown in Figs. R1(c) and R1(f)] near the singularities. Balancing the complexity, two different color bars seem to be clearer to present the physical properties (i.e., bandgaps and singularities) of the system.

Figure R5. For TE wave, phase diagrams φ_r of (a) the PC [(CGD)₂₂B]₂₀ with the rainbow color bar, and (b) the hyper-crystal E[(CGD)₂₂B]₂₀ with the blue-red color bar. The surface states support two singularities near the degenerate points: \mathcal{A} with $+1$ topological charge, together with \mathcal{B} with -1 topological charge.

(2) About **scaling**: In fact, we notice the reflection spectrum [Fig. R6(a)] and phase diagram [Fig. R6(b)] act really weird near the frequency $f = 303$ THz. This owe to the epsilon-near-zero (ENZ) property of the hyperbolic layer A, which is mimicked by subwavelength stacks as (CGD)_s. As is shown in Fig. R6(c), $\varepsilon_\perp = \varepsilon_{Az}$ marked by blue lines is close to zero near that frequency, while $\varepsilon_{//} = \varepsilon_{Ax} = \varepsilon_{Ay}$ marked by red line is not. In that case, the Bloch wave vector $k_{Az}^{TM} = \sqrt{\varepsilon_{Ax}\mu_{Ay} - \varepsilon_{Ax}k_\rho^2/\varepsilon_{Az}}$ for the TM wave is always imaginary, except the radial wave vector $k_\rho = k_0 \sin \theta$ is also zero, which means light can transmit the layer A only at norm incident (with the incident angle $\theta = 0$). However, the Bloch wave vector $k_{Az}^{TE} = k_0\sqrt{\varepsilon_{Ay}\mu_{Ax} - \mu_{Ax}k_\rho^2/\mu_{Az}}$ for the TE wave is not affected by $\varepsilon_\perp = \varepsilon_{Az}$ [Figs. 3(d) and 3(f)].

Recent years, ENZ materials have aroused heated discussion in the field of singularity and bound state in the continuum (BIC) [R3–R5]. Although ENZ physics is also intriguing, it seems to have no direct connection with our main theme “Weyl nodal line semimetal” in this manuscript. In order to better guide the readers and focus on our key idea, we reduce the range of frequency into [283, 293] THz for the TM wave.

Figure R6. (a) Reflection spectrum R and (b) phase diagram φ_r of the hyper-crystal E[(CGD)₂₂B]₂₀ for the TM wave. (c) Effective permittivity parameters of the structure (CGD)_s.

4. From Line 117 to Line 124, the paper starts to discuss the concept of “decoupled multi-quasiparticles,” which has later been further emphasized through the proposed double Weyl nodal ring structure in TE and TM modes. However, although the construction of the topological structure has been successful, the importance of this topic has not been sufficiently discussed. Is there any further photonic application or physical implication of this concept in other systems?

Response: We thank the reviewer for this constructive remark, which promotes explorations on the core concept from both application and research. Figure 1 depicts the improvement of degrees of freedom (DOFs) from single quasiparticle with Hamiltonian $H_I = H(\Delta, \phi, q_1, q_2)$ to decoupled multi-quasiparticles with Hamiltonian $H_{II} = H_I^\alpha(\Delta^\alpha, \phi^\alpha, q_1^\alpha, q_2^\alpha) \oplus H_I^\beta(\Delta^\beta, \phi^\beta, q_1^\beta, q_2^\beta)$, which might arouse some intriguing photonic applications:

(1) Inspired by Feynman’s classical thought about nanoparticles, freely modulated quasiparticles in the form of nodal rings, the simplest closed topological configurations, play the role of basic elements and provide a bottom-up method to construct more complicated band structures, like nodal chains [R6], nodal links [R7–R8], and even new topological phases. However, coupling between quasiparticles changes their band structure destructively in the process of assembling. To overcome this point, decoupled multi-quasiparticles might be necessary. We could modulate quasiparticles with different pseudospins (polarizations here) to the designed positions in the energy-momentum space, even degenerations of quasiparticles are also allowed in this process (corresponding to the Dirac Nodal phase). On the other hand, we could actively introduce coupling [R9] like gyrotropic or chiral effects [R10], which lead to more DOFs of modulation. Corresponding phenomena are reserved for further study. In the revised manuscript, we have added the following discussion into the main text (in Line 203) to elucidate the possible implications in the field of nodal physics.

“Inspired by Feynman’s classical thought about nanoparticles, quasiparticles in the form of nodal rings, the simplest closed topological configurations, may play the role of basic elements and provide a bottom-up method to construct more complicated band structures, like nodal chains, nodal links, and even new topological phases. However, coupling between quasiparticles tends to change their band structure destructively in the process of assembling. To overcome this point, decoupled multi-quasiparticles can provide a flexible platform. On the other hand, coupling [R9] like gyrotropic or chiral effects [R10] can be introduced actively, which may lead to more DOFs of modulation.”

(2) As is discussed in the main text, DSSs near the quasiparticles can support singularities. And reflection phase φ_r near singularities possesses dramatical Heaviside-like phase jumps [Fig. R1]. Such jumps are quite sensitive to environmental disturbances, such as the external refractive indexes [R4, R11], molecule attachments [R12, R13], and temperature [R14], which lays a solid foundation for sensors with ultrahigh sensitivities. To our best knowledge, these previous schemes, however, are mainly based on a single singularity with the TM component (zero points of the complex reflection ratio $\rho = r^{TM}/r^{TE}$), while the TE component contributions (pole points) tend to be neglected.

From our present results, multi-quasiparticles endow these phase jumps with decoupled properties, leading to a dual-mode sensing scheme based on both TE and TM singularities. Facing external disturbances, TE and TM modes in our systems have different but sensitive

responses, and support two times of independent measurements simultaneously. This property can infuse new blood into conventional schemes of singularity applications. For this part, more details could be found in supplementary information, Sec. VII. In the revised manuscript, we have recast the following discussion into the main text (in Line 267) to further emphasize the relevance between decoupled multi-particles and the dual-mode sensing scheme.

“Inheriting the decoupled properties from multi-quasiparticles, the different evolutions for two polarizations determine that we can review the sensing scheme in expansion space $\varphi_r^{TE} - \varphi_r^{TM}$.”

On the other hand, the effective Hamiltonian H_{II} might be also realized in the systems supporting two orthogonal modes, like polarizations elastic waves with longitudinal (P) and transverse (S) types [R15], or two different mechanisms, like electromagnetic and elastic waves [R16, R17]. In the revised manuscript, we have added the following discussion into the main text (in Line 124) to elucidate the possible physical implication in other systems.

“This effective Hamiltonian can be realized in the system supporting two orthogonal modes, like polarizations in electromagnetic or elastic waves [R15], or two different mechanisms, like electromagnetic and elastic waves [R16, R17].”

Therefore, in our opinion, “multi-quasiparticle” denotes a system possesses several special collective modes, and “decoupled” denotes these modes are independent. They provide a convenient method to modulate the energy bands, and inherit properties from the original single quasiparticles.

5. The paper eventually reached a BIC ring by reducing radiation leakages and absorption losses in its final part. Does the concept of “decoupled multi-quasiparticles” relate to the construction of the finally achieved BIC ring? In Line 265, the paper claims that the singularity pairs near the nodal line will eventually form BIC when reducing radiation leakages and absorption losses. However, is there any theoretical or calculational proof that the BIC is actually formed in this situation? Readers may not be fully convinced that there actually exists a BIC even when the singularities become very close to each other.

Response: We thank the reviewer for this illuminating comment, which promotes the understanding and description of the physical mechanisms. We fully agree with the supposition of the reviewer that the concept of “decoupled multi-quasiparticles” relates to the construction of the finally achieved BIC ring.

First, we consider there exists a pair of singularities carrying +1 and -1 topological charges in the outside and inside bandgap regions of the WNR, respectively. Singularities possess ill-defined reflection phases, namely their reflections are always zero. Hence, singularities in the bandgap can only exist on the edge states. At the same time, edge states, tunneling modes at the boundary between ENG and MNG, are restricted in the bandgap, since the silver layer E (ENG) and the bulk bands of PC (equivalent to transparent media with $\epsilon > 0$, $\mu > 0$ or $\epsilon < 0$, $\mu < 0$) can not support tunneling modes. Therefore, the moving singularities are always restricted in the bandgap in the process of band evolutions (e.g. increasing the unit cell number N). Interestingly, it might be somewhat analogous to the events with causal relationships restricted in the time-like region of the light-cone [R18]. To approach each other, the above-

mentioned singularities typically come close to the band degeneracy point (WNR). Strictly speaking, they cannot pass through the WNR, merge, and annihilate completely. Thus, there always exists a state at the WNR, namely a bound state in the continuum (BIC). From the perspective of bandwidth, we plot the reflection with fixed k_ρ/k_0 in Fig. R7. Take Dirac nodal line semimetal (DNLS) in Fig. R7(d) as an example, the linewidth of the edge state is restricted by the band edge. Therefore, it becomes narrow when coming close to the WNR (along the direction of the arrow), and ultimately vanishes in the bulk states at $k_\rho/k_0 = 0.7355$ for the WNR (yellow star), which corresponds to BIC. Similar situations could be found in the phases of quasi-DNLS [Fig. R7(e)] and isolated Weyl nodal line semimetal (WNLS) [Fig. R7(f)]. Noteworthy, the DOF of translation is endowed to BIC from the decoupled multi-quasiparticle phase of quasi-DNLS [Fig. R7(e)] to isolated WNLSs [Fig. R7(f)], which may give rise to a freely controllable BIC.

In the revised manuscript, we have added the following discussion into the supplementary information, Sec. V (in Page 22) to further emphasize the relevance between BICs and WNRs.

“To explore the properties near the BIC, we further observe the reflection with fixed k_ρ/k_0 in Fig. S11. Take Dirac nodal line semimetal (DNLS) in Fig. S11(d) as an example, the linewidth of the edge state is restricted by the band edge. Therefore, it becomes narrow and narrow when coming close to the WNR (along the direction of the arrow), and vanishes in the bulk states ultimately at $k_\rho/k_0 = 0.7355$ for the WNR (yellow star), which corresponds to BIC. Similar situations can also be found in the phase of quasi-DNLS [Fig. S11(e)] and isolated Weyl nodal line semimetal (WNLS) [Fig. S11(f)]. Note that the DOF of translation is endowed to BIC from the decoupled multi-quasiparticle phase quasi-DNLS [Fig. S11(e)] to isolated WNLSs [Fig. S11(f)], which may give rise to a freely controllable BIC.”

Worth mentioning, we notice a recent preprint discusses BICs spawned from the Dirac point (DP) in the PC slab from a different perspective of polarization defects [R19]. They find “Right at the DP, all the diffraction orders are ill-defined because of the degeneracy, indicating that the eigenstates can be mixed to any ratio to produce any amplitudes of diffraction, including zero that corresponds to the integer defect.” Wherein “the integer defect” corresponds to BIC, and “the eigenstates can be mixed to any ratio to produce any amplitudes of diffraction” can be related to any linear combination of eigenstates with different polarizations coming from decoupled multi-quasiparticles in this work. In that case, there might be a universal relevance between nodal physics and BIC. In the revised manuscript, we have cited this work in the main text (in Line 292) to reveal the universal relevance between nodal physics and BIC.

“Worth mentioning, we notice a recent work discusses BICs spawned from the Dirac point in the PC slab [R19]. They find ‘the eigenstates can be mixed to any ratio to produce any amplitudes of diffraction’, including the BIC, which may provide a different perspective to the BICs originated from decoupled multi-quasiparticles, and reveal a universal relevance between the nodal physics and the singularity physics.”

Figure R7. (FIG. S11 in the supplementary information) Reflection diagrams of bilateral DSSs for the phase of (a) DNLS [(CGD)₄₄B']_N with $E_F = 0.93$ eV, (b) quasi-DNLS [(CGD)₂₂B]_N with $E_F = 0.93$ eV, and (c) isolated WNLs [(CGD)₂₂B]_N with $E_F = 1.43$ eV. And the unit cell number is $N=200$. The second row is an enlarged view of the white box in the first row. Reflection φ_{PC} for the PC (green lines) and the hyper-crystal (purple lines) with several fixed k_ρ/k_0 for the phase of (d) DNLS, (e) quasi-DNLS, and (f) isolated WNLs. The moments of WNR are marked by yellow stars, and the bilateral DSSs are represented by blue lines. As a comparison, the bilateral DSS of quasi-DNLS is also marked as the blue dashed line in (f).

Hope with the careful revision addressing the reviewer's reports, this work could meet the high standard of *Nature Communications*.

[R1] Deng, W. M., Chen, Z. M., Li, M. Y., Guo, C. H., Tian, Z. T., Sun, K. X., Chen, X. D., Chen, W. J. & Dong, J. W. Ideal nodal rings of one-dimensional photonic crystals in the visible region. *Light Sci. Appl.* **11**, 134 (2022). <https://doi.org/10.1038/s41377-022-00821-9>.

[R2] Huang, Q. S., Guo, Z. W., Feng, J. T., Yu, C. Y., Jiang, H. T., Zhang, Z., Wang, Z. S., Chen, H. Observation of a topological edge state in the X-ray band. *Laser Photon. Rev.* **13**, 1800339 (2019). <https://doi.org/10.1002/lpor.201800339>.

- [R3] Liu, M. Q., Zhao, C., Zeng, Y. X., Chen, Y., Zhao, C. Y. & Qiu, C. W. Evolution and nonreciprocity of loss-induced topological phase singularity pairs. *Phys. Rev. Lett.* **127**, 266101 (2021). <https://doi.org/10.1103/PhysRevLett.127.266101>.
- [R4] Sakotic, Z., Krasnok, A., Alú, A. & Jankovic, N. Topological scattering singularities and embedded eigenstates for polarization control and sensing applications. *Photon. Res.* **9**, 1310 (2021). <https://doi.org/10.1364/PRJ.424247>.
- [R5] Liu, M. Q., Xia, S., Wan, W. J., Qin, J., Li, H., Zhao, C. Y., Bi, L. & Qiu, C.-W. Broadband mid-infrared non-reciprocal absorption using magnetized gradient epsilon-near-zero thin films. *Nat. Mater.* **22**, 1196 (2023). <https://doi.org/10.1038/s41563-023-01635-9>.
- [R6] Wu, Q. S., Soluyanov, A. A. & Bzdušek, T. Non-Abelian band topology in noninteracting metals. *Science* **365**, 1273–1277 (2019). <https://doi.org/10.1126/science.aau8740>.
- [R7] Yang, E. C., Yang, B., You, O. B., Chan, H. C., Mao, P., Guo, Q. H., Ma, S. J., Xia, L. B., Fan, D. Y., Xiang, Y. J. & Zhang, S. Observation of non-Abelian nodal links in photonics. *Phys. Rev. Lett.* **125**, 033901 (2020). <https://doi.org/10.1103/PhysRevLett.125.033901>.
- [R8] Wang, D. Y., Yang, B., Guo, Q. H., Zhang, R. Y., Xia, L. B., Su, X. Q., Chen, W. J., Han, J. G., Zhang, S. & Chan, C. T. Intrinsic in-plane nodal chain and generalized quaternion charge protected nodal link in photonics. *Light Sci. Appl.* **10**, 83 (2021). <https://doi.org/10.1038/s41377-021-00523-8>.
- [R9] Kim, M., Jacob, Z. & Rho, J. Recent advances in 2D, 3D and higher-order topological photonics. *Light Sci. Appl.* **9**, 130 (2020). <https://doi.org/10.1038/s41377-020-0331-y>.
- [R10] Hou, J. P., Li, Z. T., Luo, X.-W., Gu, Q. & Zhang, C. W. Topological bands and triply degenerate points in non-Hermitian hyperbolic metamaterials. *Phys. Rev. Lett.* **124**, 073603 (2020). <https://doi.org/10.1103/PhysRevLett.124.073603>.
- [R11] Ermolaev, G., Voronin, K., Baranov, D. G., Kravets, V., Tselikov, G., Stebunov, Y., Yakubovsky, D., Novikov, S., Vyshnevyy, A., Mazitov, A., Kruglov, I., Zhukov, S., Romanov, R., Markeev, A. M., Arsenin, A., Novoselov, K. S., Grigorenko, A. N. & Volkov, V. Topological phase singularities in atomically thin high-refractive-index materials. *Nat. Commun.* **13**, 2049 (2022). <https://doi.org/10.1038/s41467-022-29716-4>.
- [R12] Kravets, V. G., Schedin, F., Jalil, R., Britnell, L., Gorbachev, R. V., Ansell, D., Thackray, B., Novoselov, K. S., Geim, A. K., Kabashin, A. V. & Grigorenko, A. N. Singular phase nano-optics in plasmonic metamaterials for label-free single-molecule detection. *Nat. Mater.* **12**, 304 (2013). <https://doi.org/10.1038/NMAT3537>.
- [R13] Sreekanth, K. V., Sreejith, S., Han, S., Mishra, A., Chen, X. X., Sun, H. D., Lim, C. T. & Singh, R. Biosensing with the singular phase of an ultrathin metal-dielectric nanophotonic cavity. *Nat. Commun.* **9**, 369 (2018). <https://doi.org/10.1038/s41467-018-02860-6>.
- [R14] Tsurimaki, Y., Tong, J. K., Boriskina, S. V., Semenov, A., Ayzatsky, M. I., Machekhin, Y. P., Chen, G. & Boriskina, S. V. Topological engineering of interfacial optical Tamm states for highly sensitive near-singular-phase optical detection. *ACS Photon.* **5**, 929–938 (2018). <https://doi.org/10.1021/acsp Photonics.7b01176>.
- [R15] Liu, F. M. & Liu, Z. Y. Elastic waves scattering without conversion in metamaterials with simultaneous zero indices for longitudinal and transverse waves. *Phys. Rev. Lett.* **115**, 175502 (2015). <https://doi.org/10.1103/PhysRevLett.115.175502>.
- [R16] Maldovana, M. & Thomas, E. L. Simultaneous localization of photons and phonons in two-dimensional periodic structures. *Appl. Phys. Lett.* **88**, 251907 (2006).

<https://doi.org/10.1063/1.2216885>.

[R17] Ma, T.-X., Liu, J., Zhang, C. Z. & Wang, Y.-S. Topological edge and interface states in phoxonic crystal cavity chains. *Phys. Rev. A* **106**, 043504 (2022).

<https://doi.org/10.1103/PhysRevA.106.043504>.

[R18] Huang, H. Q., Jin, K.-H. & Liu, F. Black hole horizon in a Dirac semimetal $\text{Zn}_2\text{In}_2\text{S}_5$. *Phys. Rev. B* **98**, 121110 (2018). <https://doi.org/10.1103/PhysRevB.98.121110>.

[R19] Yin, X. F., Inoue, T., Peng, C. & Noda, S. Origins and conservation of topological polarization defects in resonant photonic-crystal diffraction. arXiv:2310.20336 [physics.optics] (2023). <https://doi.org/10.48550/arXiv.2310.20336>.

Reviewer #3 (Remarks to the Author):

In the manuscript entitled “Hyperbolic Metamaterial Empowered Controllable Photonic Weyl Nodal Line Semimetals”, Hu et al. reported the realization of quasi-DNLS, isolated WNLS, etc. by using a 1D hyper-crystal containing hyperbolic metamaterials. Especially, for the TM modes, phase transitions between three types of WNLS can be achieved through the change of the ratio between the thickness of layer A and layer B. Then, by introducing a silver layer at the surface, the structure can support drumhead surface state (DSS) with phase singularities for both TE and TM polarizations. The generated phase singularities will approach each other and eventually merge with the increase of the number of the unit cell.

Response: We thank the reviewer for his/her careful reading and positive assessments of our manuscript. We also thank this reviewer for raising the following important comments, and for indicating explicitly the ways to improve the presentation of this manuscript.

This manuscript presents good theoretical analyzations and convincing numerical simulations. I believe that this idea proposed by the authors is interesting and provides a new approach to realize varieties of topological effects in photonic systems. I would like to recommend the publication of this manuscript in Nature Communications.

Response: We thank the reviewer for recommending our paper to be published in *Nature Communications*. In particular, the reviewer considers that "This manuscript presents good theoretical analyzations and convincing numerical simulations. I believe that this idea proposed by the authors is interesting and provides a new approach to realize varieties of topological effects in photonic systems". Based on the reviewer's excellent suggestions, we have carefully addressed all the comments in the revised paper. The following is a point-to-point response to the reviewer's comments.

Some minor points:

1. Is there any difference of the phase singularities in Figure 3 for different thicknesses of the sliver layer E?

Response: We thank the reviewer for this constructive comment, which promotes a convenient method for phase singularity modulation. In fact, the phase singularities will move in the $f - k_\rho$ space, and even annihilate each other when altering the thickness of the silver layer E. With an increase of d_E from 15 nm to 20 nm to 25 nm, the reduced reflection phases at $k_\rho/k_0 = 0.8$ are highlighted on the cylinder by dark, moderate, and light red lines for both TE [Fig. R8(a)] and TM [Fig. R8(b)] waves. Such reductions lead to blueshifts of the edge state and movements of relevant singularities [Figs. R8(c)–(d)]. In this process, pairs of singularities originating from a common WNR, like \mathcal{A} & \mathcal{B} or \mathcal{C} & \mathcal{D} , come away from each other. Especially, tracing the evolution trajectory of +1 singularity \mathcal{A} (red dotted lines), it approaches -1 singularity \mathcal{E} (blue dotted lines), and vanishes after a collision with $d_E = 21.6$ nm approximately. As is shown in Fig. R8(e), this annihilation happens near the point $f_0 = 293.31$ THz, $k_{\rho 0}/k_0 = 0.81$ (black cross), which verifies clearly the conservation of topological charges in this system.

In order to better guide the readers, we have added the following discussion into the supplementary information, Sec. VIII (in Page 31) to emphasize this method.

“In addition, a convenient method to realize similar singularity annihilations is through altering the thickness d_E of the silver layer E. With an increase of d_E from 15 nm to 20 nm to 25 nm, the reduced reflection phases at $k_\rho/k_0 = 0.8$ are highlighted on the cylinder by dark, moderate, and light red lines for both TE [Figs. S16(a)] and TM [Figs. S16(b)] waves. Such reductions lead to blueshifts of the edge states and movements of relevant singularities [Figs. S16(c)–(d)]. In this process, pairs of singularities originating from a common WNR, like \mathcal{A} & \mathcal{B} or \mathcal{C} & \mathcal{D} , come away from each other. Tracing the evolution trajectory of +1 singularity \mathcal{A} (red dotted lines), it approaches -1 singularity \mathcal{E} (blue dotted lines), and vanishes after a collision with $d_E = 21.6$ nm approximately. As is shown in Fig. S16(e), this annihilation happens near the point $f_0 = 293.31$ THz, $k_{\rho 0}/k_0 = 0.81$ (black cross), which verifies the conservation of topological charges in this system.”

Figure R8. (FIG. S16 in the revised supplementary information) Topological charge annihilation by altering the thickness of the silver layer d_E . When $k_\rho/k_0 = 0.8$ for the TE wave (a) and TM wave (b), the reflection phases φ_{Ag} for the silver layer E (red), φ_{PC} for the PC [(CGD)₂₂B]₂₀ (blue) together with their sum $\varphi_{Ag} + \varphi_{PC}$ (magenta) are illustrated on the cylinders, and related reflections R_{PC} (green) together with R_{Ag+PC} (orange) are illustrated in the insets. Their colors are dark/moderate/bright for the cases of different thickness of the silver layer $d_E = 15/20/25$ nm, respectively. The boundary of $\varphi_r = \pi(-\pi)$ and the center of $\varphi_r = 0$ are highlighted by black and white dotted lines, respectively. Points that meet the condition of $\varphi_{Ag} + \varphi_{PC} = 0$ are marked by cyan stars. For the TE wave (c) and TM wave (d), vertical slice figures are the reflection spectra with different d_E . The trajectories for \mathcal{A} and \mathcal{C} with +1 topological charge, and \mathcal{B} , \mathcal{D} and \mathcal{E} with -1 topological charge are highlighted by the red and blue dotted lines, respectively. (e) Detailed evolution trajectories of these singularities for $d_E \in [1, 30]$ nm. Several momentous points corresponding to $d_E = 5/10/15/20/25$ nm are marked by circles (TE) or triangles (TM) with the color red (+1 singularities) and blue (-1 singularities), respectively. The point of annihilation between \mathcal{A} and \mathcal{E} is marked by the black cross. The inset illustrates the situations for the TM wave with an enlarged view.

2. It's interesting that the phase singularities will evolve with the increasing number of the unit cell, and the quality factor will also increase to an ultrahigh value. For traditional BICs, they usually correspond to modes with infinite quality factor. Here, I may suggest the high-Q modes in this manuscript to be called quasi-BICs.

Response: We thank the reviewer for this constructive remark. We fully agree with the reviewer that “quasi-BIC” is an appropriate description here. Based on the reviewer's very good suggestions, we have modified the related descriptions in the revised manuscript.

Hope with the careful revision addressing the reviewer's reports, this work could meet the high standard of *Nature Communications*.

REVIEWERS' COMMENTS

Reviewer #2 (Remarks to the Author):

The authors have addressed all my concerns raised in the last report. I have no further questions.

Reviewer #3 (Remarks to the Author):

In the revised manuscript, the authors have provided suitable responses to address all my questions. And I recommend the publication in Nature Communications.